# Pyramidal Recursive Composition of Multi-Word Units into Unified Representations

## Abstract

In this paper, we explore the composition of word embeddings to create richer, more meaningful representations of multi-word units. Existing methods, such as averaging word embeddings, provide simple and efficient approaches. However, they often fail to capture the complexity of multi-word interactions. To address this, we employ the Pyramidal Recursive learning (PyRv) method, which recursively combines word embeddings into unified representations. Originally developed for constructing representations hierarchically from subwords to phrases, PyRv is well-suited for progressively merging individual word vectors into phrase vectors. We evaluate the effectiveness of PyRv for embedding composition using fastText embeddings on the dependency relation labeling task. Using a single fastText word embedding yields an accuracy of 71%. Averaging five fastText word embeddings (the middle word and its four neighboring words) results in a significant drop in accuracy to 34%. In contrast, by composing five word embeddings with PyRv, we achieve an accuracy of 77%, demonstrating the superior ability of PyRv to integrate multiple word embeddings into more expressive representations. These findings highlight the potential of PyRv as a lightweight yet powerful technique for word embedding composition.

## 1 Introduction

Word embeddings are foundational to many natural language processing (NLP) tasks, providing a way to map words into continuous vector spaces that capture semantic relationships between them. By converting words into numerical representations, word embeddings allow machines to process and understand text in a way that retains important linguistic properties. Popular word embedding methods, such as Word2Vec (Mikolov et al., 2013), GloVe (Pennington et al., 2014), and FastText (Bojanowski et al., 2017), have demonstrated the utility of these representations by positioning semantically similar words closer in the vector space. The effectiveness of NLP models often hinges on the quality of these embeddings, as richer and more informative representations can lead to improved performance across various downstream tasks.

In many real-world applications, however, understanding text at the word level alone is insufficient. The need to represent larger linguistic units, such as phrases or sentences, necessitates techniques for combining word embeddings into more complex structures. Word composition, which involves aggregating individual word embeddings to represent multi-word expressions or entire sentences, serves this purpose. By integrating information from multiple word vectors, compositional methods aim to capture both the meanings of individual words and the syntactic and semantic interactions between them, particularly in morphologically complex languages, such as Croatian, which was used for evaluation in this paper.

There are several established methods for combining word embeddings. Simple techniques include element-wise operations such as addition, averaging, or multiplication, which produce a composite vector by leveraging individual word vectors (such as in Joulin et al. (2016) and Arora et al. (2017)). These methods, while computationally efficient, may fail to fully capture the complexity of phrase or sentence meaning. More sophisticated approaches employ weighted combinations, context-aware methods, or syntactic structures to improve the expressiveness of the resultant embeddings (such as in Socher et al. (2013) and Bahdanau (2014)).

While transformer-based models, such as BERT (Devlin et al., 2018) and GPT (Brown et al., 2020), offer robust pre-trained sentence embeddings by learning deep contextual representations, they are often computationally expensive and may not always align with the specific needs of certain tasks. Word composition methods provide a more lightweight and flexible alternative, especially in cases where transparency and control over the aggregation process are crucial. Additionally, word-level composition techniques can better retain the granularity of individual word meanings, which is sometimes diluted in sentence-level embeddings produced by transformer models.

Word composition provides a valuable approach for constructing meaningful representations of multi-word units, balancing computational efficiency and interpretability. These methods remain relevant, particularly in domains where sentence embedding techniques may obscure important details or where domain-specific customization of embedding composition is required.

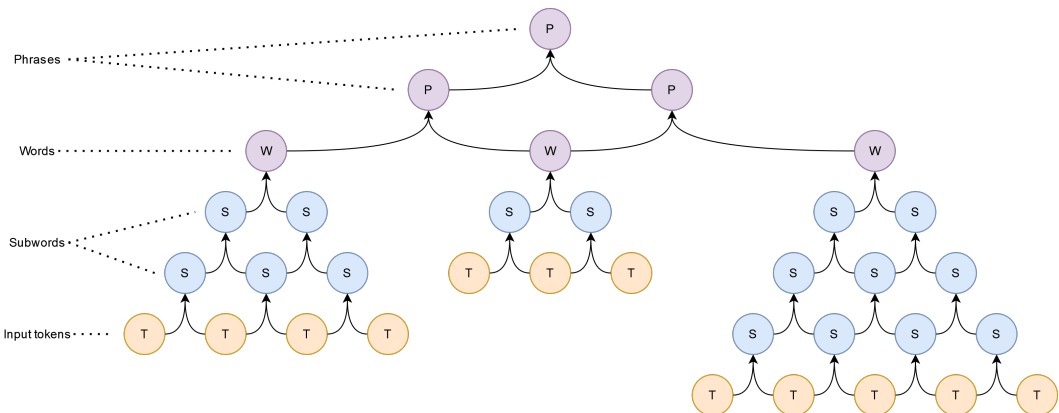

Figure 1: A visualized example of a pyramidal recursion in the PyRv method. The lowest-level nodes correspond to input tokens. Moving upward, the nodes within the three pyramids represent combined subword embeddings. At the pyramid peaks, nodes represent word embeddings, and higher nodes signify combined word embeddings, representing phrases.

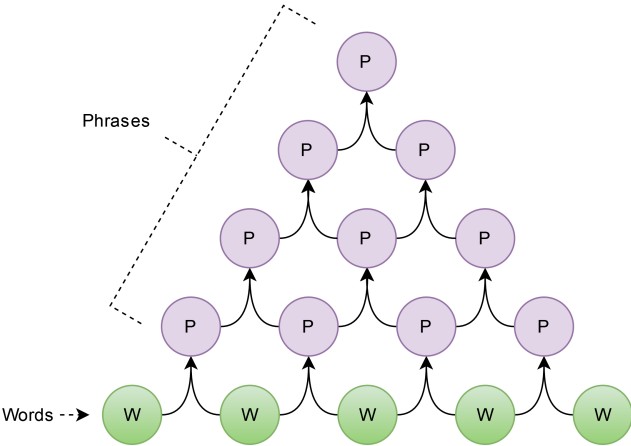

Figure 2: A visual representation of pyramidal recursion in the PyRv+FT method. The lowest-level nodes correspond to fastText-embedded words. As we move upward, the nodes represent combined word embeddings, capturing phrase-level meanings.

In this work, we leverage the Pyramidal Recursive learning (PyRv) method, introduced in Babić & Meštrović (2024), to compose multiple word embeddings into unified representations. PyRv facilitates structured composition through its hierarchical learning approach, recursively combining representations at each level of abstraction. Initially developed for constructing representations from tokens (subwords) up to phrases (as illustrated in Figure 1), PyRv is well-suited for combining word

embeddings by progressively merging individual word vectors into phrase vectors (as is shown in Figure 2).

One of the key properties of the PyRv method is representation compositionality, which enables the composition of multiple embeddings into a coherent, semantically rich representation. This property aligns with the objective of this paper, where the focus is on effectively combining word embeddings to capture more complex linguistic structures. By recursively merging embeddings, PyRv maintains the semantic integrity of each word while constructing increasingly abstract representations at higher levels of the hierarchy.

The primary contribution of this paper is the introduction of a method for composing multi-word units into unified representations using Pyramidal Recursive learning (PyRv). To assess the effectiveness of this approach, we train the PyRv embedding model on Croatian texts and compare its performance to the widely-used baseline method of averaging word embeddings. In addition, we explore the structure of the representation space generated by PyRv's composition method. Our evaluation results validate PyRv's compositionality property.

Following this introduction, the subsequent sections of this paper are structured as follows: In Section 2, "Related Work," we explore prior research, highlighting methods that compose word embeddings. Section 3, "Embedding Method," presents our method for composition of word embeddings. Section 4, "Evaluation," details the datasets, experiment setup, and results. Finally, in Section 5, "Conclusion," we conclude with a summary of our contributions to the field and discuss future directions.

## 2 RELATED WORK

Composing word embeddings to generate meaningful representations of larger text units remains a critical area of study in natural language processing. Approaches to this problem span from simple aggregation techniques that compose word embeddings to more complex neural architectures that embed entire sentences, each offering unique advantages in different contexts.

A common approach is to average word embeddings to generate a single vector representing a phrase or sentence. Joulin et al. (2016) introduced this idea in the context of fastText, where word embeddings are averaged and subsequently used for efficient text classification. This technique, inspired by the continuous bag of words (CBOW) model (Mikolov et al., 2013), offers a computationally lightweight solution that performs competitively with deeper models in various NLP tasks.

Building upon this, Arora et al. (2017) proposed an enhanced version where word embeddings are combined using weighted averages, followed by post-processing through PCA/SVD. The weighting scheme they propose significantly improves performance on textual similarity tasks. This method demonstrates that simple compositional techniques can rival more complex architectures, especially in unsupervised settings.

Wieting et al. (2015) conducted a comparative study that highlighted the trade-offs between simple word averaging and more complex models like LSTMs for sentence embedding. Their findings showed that while LSTMs perform well on in-domain data, simple word averaging techniques tend to outperform LSTMs in out-of-domain tasks. This suggests that straightforward compositional methods, despite their simplicity, are robust and generalizable across diverse datasets.

Recursive models have also been explored for word compositionality. Socher et al. (2013) proposed a recursive neural network based on syntactic parse trees to generate phrase and sentence representations, useful in tasks like sentiment analysis. Zhao et al. (2015) introduced a Self-Adaptive Hierarchical Sentence Model, using recursive structures without relying on syntax, showing that supervised learning can effectively capture compositional semantics in a non-syntactic hierarchy.

Transformer (Vaswani, 2017) architectures use attention mechanism to compose embeddings. Bahdanau (2014) introduced the attention mechanism in neural machine translation, allowing models to dynamically focus on different parts of the input sequence during decoding. The introduction of attention helped relieve the encoder from compressing all information into a single fixed-length vector, thus enabling more flexible and effective composition of representations over sequential data.

In this work, we build on these approaches by applying the Pyramidal Recursive learning (PyRv) method (Babić & Meštrović, 2024) to recursively combine word embeddings into more abstract representations. Unlike the averaging techniques of Joulin et al. (2016) and Arora et al. (2017), PyRv enables hierarchical composition, capturing both word-level and higher-level semantic structures in a more structured way. Unlike most other recursive models, such as the one introduced by Socher et al. (2013), which rely on syntactic trees, PyRv operates without requiring explicit parse structures, and unlike the model introduced by Zhao et al. (2015), it is fully unsupervised. Additionally, by recursively merging embeddings, our approach offers a simpler alternative to attention mechanisms for constructing rich representations of text.

## 3 EMBEDDING METHOD

In this section, we introduce word embeddings and common composition techniques, followed by a detailed explanation of how the PyRv method is applied to compose word embeddings.

### 3.1 WORD EMBEDDING

Word embeddings, such as those produced by fastText (Bojanowski et al., 2017), are commonly used to represent individual words. To represent multiple words, basic techniques like averaging or concatenation can be applied. However, both approaches come with notable limitations.

When averaging embeddings (e.g., taking the mean of multiple word vectors), important information, particularly word order, is lost. Concatenation preserves all information but presents two significant challenges.

First, concatenated embeddings vary in dimensionality depending on the number of words, which complicates their use as input for models that require a fixed input size. Second, the dimensionality of concatenated representations can grow excessively large. For instance, a 5-word phrase embedded with 300-dimensional fastText results in a 1500-dimensional vector (5x300).

In our evaluation, we use averaging to compose Croatian fastText embeddings (Grave et al., 2018) to maintain a consistent dimensionality across all methods, ensuring that the results are comparable.

### 3.2 PYRV WITH FASTTEXT WORD EMBEDDINGS

Pyramidal Recursive learning (PyRv) is a method designed to construct hierarchical representations of text, moving progressively from low-level units such as characters or subwords to higher-level representations such as words, phrases, sentences, and even paragraphs. PyRv combines representations recursively, forming increasingly abstract and semantically rich embeddings at each level of the hierarchy.

To address the limitations of averaging and concatenation, we explore the use of PyRv for composing multiple word embeddings into a single, unified representation. Unlike previous work on PyRv (Babić & Meštrović, 2024), where the recursion starts from subwords or tokens, in this study we begin with word embeddings produced by fastText. This hybrid approach is referred to as PyRv+FT.

For this paper, we use pre-trained Croatian fastText word vectors (Grave et al., 2018) to embed words, which are then recursively combined into phrase embeddings via the PyRvNN model. The PyRv+FT embeddings are trained on Croatian Wikipedia texts (Wikimedia public dump, January 11, 2020) for 10 epochs.

We evaluate PyRv+FT on two downstream tasks, described in detail in the next section. Through this process, we investigate how PyRv improves the compositionality of word embeddings, while maintaining manageable dimensionality and enhancing representational quality.

## 4 EVALUATION

In this section, we describe the evaluation of fastText and PyRv+FT embeddings on two key NLP tasks: Universal Part-of-Speech tagging (UPOS) and Universal Dependency Relation labeling (DE-PREL). UPOS tags represent core grammatical categories such as nouns, verbs, and adjectives, while

DEPREL captures the syntactic relationships between words in a sentence, indicating dependencies like subjects, objects, and modifiers.

For this evaluation, we use the hr500k 2.0 dataset (Ljubešić et al., 2016), a Croatian corpus with labeled data for both UPOS and DEPREL tasks (amongst others). This dataset contains 901 texts, 24,763 sentences, and a total of 499,635 tokens.

Additionally, we perform a qualitative analysis of PyRv+FT embeddings by visualizing the representation space to gain deeper insights into its structure and characteristics.

## 4.1 EXPERIMENT SETUP

To assess the performance of different embedding methods on the downstream tasks of UPOS and DEPREL, we use a multi-layer perceptron (MLP) model with one hidden layer. The hidden layer consists of 1,000 neurons and uses the ReLU activation function. The input to the model is a 300-dimensional vector (the size of both fastText and PyRv+FT embeddings). The output layer, with softmax activation, adjusts to the number of classes in each task: 17 classes for UPOS and 37 classes for DEPREL. Each evaluation is conducted by training the MLP for one epoch.

The embedding procedure remains consistent across both UPOS and DEPREL tasks, differing only in the MLP output labels.

The method of embedding a word from a sentence depends on the embedding strategy employed:

- **fastText 1 word**: Embeds only the target word, ignoring its surrounding context.
- **mean fastText N words**: Represents the target word by embedding all N words (the target word and its N-1 neighboring context words) separately and averaging the embeddings to obtain a final representation.
- **PyRv+FT N words**: Embeds each word using fastText, but instead of averaging the N embeddings, it recursively combines them using the PyRvNN model to generate a single, unified embedding.

## 4.2 QUANTITATIVE RESULTS

Table 1: UPOS results, Macro and Weighted averages.

| | Accuracy | Precision | | Recall | | F1 score | |
|---|---|---|---|---|---|---|---|
| | | M. avg | W. avg | M. avg | W. avg | M. avg | W. avg |
| fastText 1 word | **0.95** | **0.91** | **0.95** | **0.89** | **0.95** | 0.89 | **0.95** |
| mean fastText 3 words | 0.61 | 0.57 | 0.61 | 0.59 | 0.61 | 0.57 | 0.61 |
| PyRv+FT 3 words | 0.93 | 0.9 | 0.93 | **0.89** | 0.93 | **0.9** | 0.93 |

Table 2: DEPREL results, Macro and Weighted averages.

| | Accuracy | Precision | | Recall | | F1 score | |
|---|---|---|---|---|---|---|---|
| | | M. avg | W. avg | M. avg | W. avg | M. avg | W. avg |
| fastText 1 word | 0.71 | 0.52 | 0.68 | 0.48 | 0.71 | 0.47 | 0.68 |
| mean fastText 5 words | 0.34 | 0.25 | 0.34 | 0.19 | 0.34 | 0.19 | 0.31 |
| PyRv+FT 5 words | **0.77** | **0.58** | **0.77** | **0.55** | **0.77** | **0.56** | **0.76** |

**UPOS.** Part-of-speech tagging is a relatively simple task where the surrounding word context does not provide significant benefits for classification. We include UPOS evaluation primarily to demonstrate how averaging fastText word embeddings can degrade downstream performance, while combining fastText word embeddings using PyRvNN preserves much of the embedding quality.

When using fastText to embed a single word without considering its context, we achieve an accuracy of 95%. However, averaging fastText embeddings over three words leads to a substantial drop in

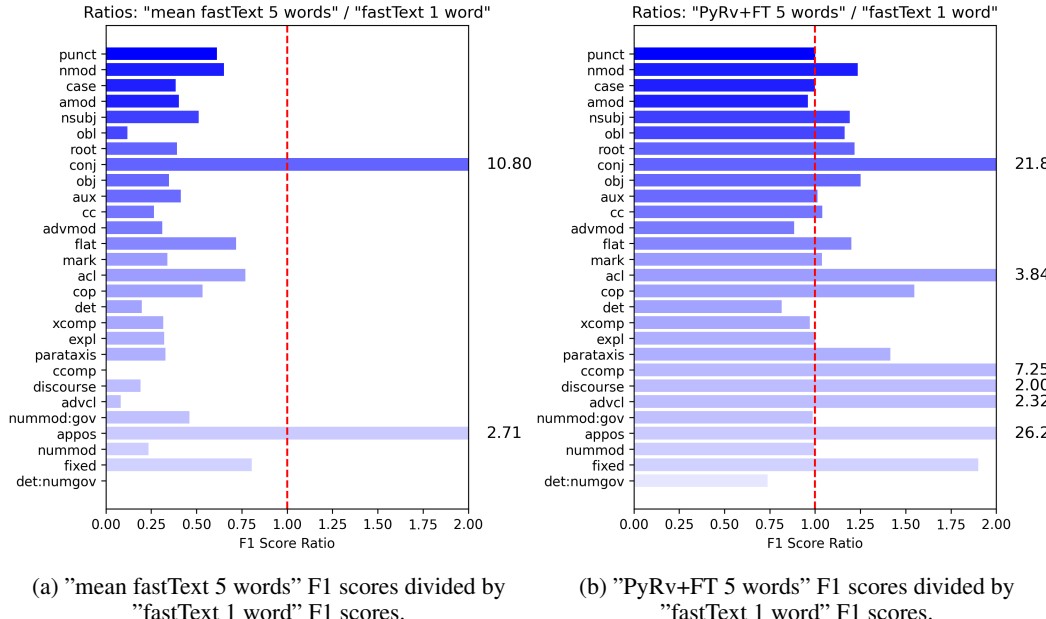

(a) "mean fastText 5 words" F1 scores divided by "fastText 1 word" F1 scores.

(b) "PyRv+FT 5 words" F1 scores divided by "fastText 1 word" F1 scores.

Figure 3: DEPREL relative F1 score ratios (by class) comparing different composition methods. The left plot (a) shows the ratio of F1 scores for "mean fastText 5 words" versus "fastText 1 word", while the right plot (b) compares "PyRv+FT 5 words" versus "fastText 1 word". Each bar represents a class, and the length of the bar indicates the relative performance of the model. Classes are ordered by support value in the test set (larger on the top).

performance, with accuracy falling to 61%. In contrast, when combining fastText embeddings for three words using PyRvNN, the performance degradation is minimized, yielding an accuracy of 93%. These results highlight how PyRvNN can effectively mitigate the loss of information that occurs when averaging word embeddings. Detailed results are shown in Table 1.

**DEPREL.** The dependency relation task is more complex than UPOS, as it requires understanding the syntactic relationships between words. In this case, enriching word embeddings with surrounding context can significantly improve classification performance.

When embedding a single word using fastText (without its context), the model achieves an accuracy of 71%. However, averaging five fastText word embeddings, including the target word and its four neighbors, results in a sharp decline in performance, with accuracy dropping to 34%. This reduction in accuracy reflects how averaging word embeddings leads to the loss of important information, including word order and syntactic structure. By contrast, composing five fastText word embeddings using PyRvNN boosts accuracy to 77%, demonstrating the method's ability to capture more nuanced relationships between words.

Evaluation results are presented in Table 2 with more detailed results (by class) available in Tables 5, 6, and 7 in the Appendix.

Figure 3 presents bar plots comparing F1 scores between two composition methods, mean averaging and PyRv, relative to single word fastText embeddings' performance. The left plot 3a illustrates the F1 score ratio between "mean fastText 5 words" and "fastText 1 word", while the right plot 3b contrasts "PyRv+FT 5 words" with "fastText 1 word".

In the following analysis, we focus on three notable classes: punctuation, conjuncts, and adnominal clauses. These were selected because punctuation classification does not rely on context, classifying conjuncts without context is nearly impossible, and for adnominal clauses, context is helpful but averaging tends to degrade performance.

**Punctuation** (punct) refers to punctuation marks such as ".", "?", "!", and ",". Since punctuation is straightforward to classify without context, a single fastText embedding for the target word alone

achieves a perfect F1 score of 1. Averaging five fastText embeddings (the target word and its four neighboring words) significantly degrades performance, reducing the F1 score to 0.61. However, using PyRv to compose these embeddings preserves the high performance, maintaining an F1 score of 1.

**Conjunct** (conj) denotes a relation between elements connected by coordinating conjunctions like "and," "or," or ",". In coordinate structures, the first element is conventionally treated as the head, with subsequent elements connected through the conj relation. For example, in the sentence "Bill is *big* and *honest*," the word "honest" is labeled as conj (connected to "big"). Similarly, in "He *came* home, *took* a shower and immediately *went* to bed," the words "took" and "went" are both labeled as conj (connected to "came"). Classifying conjuncts accurately requires contextual information. A single fastText word embedding, without any context, yields a poor F1 score of 0.03. Averaging the embeddings of the target word and its four neighbors improves performance significantly, achieving an F1 score of 0.32 (a 10.8x improvement). Composing these embeddings using PyRv further boosts performance, reaching an F1 score of 0.64 (a 21.85x improvement over the single word embedding).

**Adnominal clause** (acl) refers to finite or non-finite clauses that modify a nominal. For instance, in "the *issues* as he *sees* them," the word "sees" is labeled as acl (connected to "issues"). In "There are many online *sites offering* booking facilities," the word "offering" is labeled as acl (connected to "sites"). Using a single fastText word embedding results in an F1 score of 0.14. Averaging the target word's embedding with its four neighbors actually degrades performance slightly, reducing the F1 score to 0.11. In contrast, composing these word embeddings with PyRv substantially improves performance, raising the F1 score to 0.53 (a 3.84x improvement over the single word embedding).

### 4.3 QUALITATIVE ANALYSIS

To gain qualitative insights into the structure of PyRv+FT embeddings, we visualize the representation space. A portion of this space is shown in Figures 4 and 5 (in the Appendix). In these visualizations, each node represents a phrase consisting of two or three words. A two-word phrase is connected to a three-word phrase if the shorter phrase is part of the longer one.

We highlight specific clusters within the visualized space, with detailed examples of phrases from these clusters (translated to English) provided in Tables 3 and 4 (Tables with original phrases in Croatian are in Appendix: 8 and 9). The areas circled in the figures contain phrases built around the prepositions "u" (Croatian for "in") and "na" (Croatian for "on"). For example, Area C contains two-word phrases like "primjena **na**" (eng. "application on"), while Area A includes phrases such as "**na** svijet" (eng. "on the world"). Similarly, Area B features three-word phrases like "primjena **na** svijet" (eng. "application on the world"), and Area D includes phrases like "**na** svijet oko" (eng. "on the world around"). Same holds for the preposition "u".

The organization of phrases in the representation space is not random: phrases with similar syntactic structures (e.g., where the preposition appears at the beginning, middle, or end of the phrase) tend to cluster together. Furthermore, within these broader areas, smaller sub-clusters form based on the specific preposition ("u" or "na") present in the phrase.

## 5 CONCLUSION

In this paper, we explored the use of Pyramidal Recursive learning (PyRv) for the composition of word embeddings and evaluated its ability to generate meaningful representations of multi-word units. Our findings show that PyRv outperforms simple averaging methods in embedding composition.

In the part-of-speech tagging task, where word context is less crucial, single word embeddings achieve an accuracy of 95%. Averaging 3-word context embeddings reduces this to 61% due to the loss of word order information, while PyRv retains a high accuracy of 93% by effectively preserving word order. In the more complex task of dependency relation labeling, where single word embeddings reach 71% accuracy, averaging embeddings for 5-word contexts results in a sharp decline to 34%. In contrast, composing context words with PyRv attains a significantly higher accuracy of 77%, demonstrating its superior capability in integrating multiple word embeddings into cohesive

Table 3: Phrases by areas (A, B, and C) in the PyRv+FT representation space, translated to English (some phrases are longer when translated).

| Preposition "on" (cro. "na") | | |
| --- | --- | --- |
| **Area C** | **Area B** | **Area A** |
| application on | application on the world | on the world |
| are on | are on local | on local |
| assistant on | assistant on the subject | on the subject |
| media on | media on protest | on the protest |
| vat on | vat on tickets | on tickets |
| based on | based on data | on data |
| finance on | finance on revenues | on revenues |
| os on | os on which | on which |
| dollars on | dollars on google | on google |
| found on | found on the third | on the third |
| relation on | relation on the past | on the past |

| Preposition "in" (cro. "u") | | |
| --- | --- | --- |
| **Area C** | **Area B** | **Area A** |
| enthusiast in | enthusiast in the river | in the river |
| currently in | currently in testing | in testing |
| released in | released in circulation | in circulation |
| circulation in | circulation in June | in June |
| by activity in | by activity in teaching | in teaching |
| work in | work in the wood | in the wood |
| tickets in | tickets in europe | in europe |
| musician in | musician in croatia | in croatia |
| only in | only in the past | in the past |
| drop in | drop in the sea | in the sea |
| is in | is in the past | in the past |
| year in | year in croatia | in croatia |
| percent in | percent in relation | in relation |
| and in | and in the average | in the average |
| . in | . in this | in this |

Table 4: Phrases in area D in the PyRv+FT representation space, translated to English (some phrases are longer when translated).

| Area D | |
| --- | --- |
| **Preposition "on" (cro. "na")** | **Preposition "in" (cro. "u")** |
| on the world around | in the river which |
| on local elections | in testing and |
| on the subject of organization | in circulation in |
| on the protest of musicians | in june 2009 |
| on tickets for | in teaching 1982 |
| on the data collected | in the wood industry |
| on budget revenues | in europe . |
| on which this | in croatia only |
| on google play | in the past two |
| on the third position | in the sea of state |
| on the past year | in the past year |
| | in croatia is not |
| | in relation to |
| | in the average spends |
| | in this praise |

and expressive representations. This validates the effectiveness of the compositionality property of PyRv in real-world tasks.

The primary contribution of this work is the introduction of a method for composing multi-word units into unified representations using PyRv. We provided an evaluation of its effectiveness compared to averaging word embeddings and validated its compositionality property. Additionally, we explored the structure of the representation space generated by PyRv's compositional approach. By training the PyRv model on Croatian texts, we demonstrated its flexibility and potential for application across diverse languages.

Looking ahead, future work could include expanding the evaluation of PyRv to other NLP tasks, beyond the UPOS and DEPREL tasks, and it could include comparison with more composition methods. Investigating different PyRv architectures also presents an exciting opportunity for future research.

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

# A   APPENDIX

Table 5: DEPREL evaluation results using the fastText embedding method (single-word embeddings).

| Class | Precision | Recall | F1 score | Support |
|---|---|---|---|---|
| punct | 1 | 1 | 1 | 3037 |
| nmod | 0.6 | 0.58 | 0.59 | 2437 |
| case | 0.96 | 0.98 | 0.97 | 2364 |
| amod | 0.79 | 0.93 | 0.85 | 2355 |
| nsubj | 0.53 | 0.66 | 0.59 | 1725 |
| obl | 0.48 | 0.57 | 0.52 | 1607 |
| root | 0.45 | 0.67 | 0.53 | 1136 |
| conj | 0.19 | 0.02 | 0.03 | 1134 |
| obj | 0.49 | 0.44 | 0.46 | 1072 |
| aux | 0.75 | 0.96 | 0.84 | 1037 |
| cc | 0.83 | 0.97 | 0.89 | 887 |
| advmod | 0.8 | 0.88 | 0.84 | 825 |
| flat | 0.54 | 0.69 | 0.61 | 689 |
| mark | 0.85 | 0.91 | 0.88 | 471 |
| acl | 0.44 | 0.08 | 0.14 | 452 |
| cop | 0.58 | 0.22 | 0.32 | 415 |
| det | 0.87 | 0.87 | 0.87 | 400 |
| xcomp | 0.61 | 0.7 | 0.65 | 350 |
| expl | 0.85 | 1 | 0.92 | 302 |
| parataxis | 0.63 | 0.38 | 0.47 | 300 |
| ccomp | 0.18 | 0.02 | 0.03 | 230 |
| discourse | 0.48 | 0.21 | 0.29 | 208 |
| advcl | 0.29 | 0.08 | 0.12 | 198 |
| nummod:gov | 0.72 | 0.9 | 0.8 | 187 |
| appos | 1 | 0.01 | 0.02 | 130 |
| nummod | 0.82 | 0.63 | 0.71 | 117 |
| fixed | 0.34 | 0.33 | 0.34 | 100 |
| csubj | 0 | 0 | 0 | 40 |
| det:numgov | 0.73 | 0.66 | 0.69 | 29 |
| orphan | 0 | 0 | 0 | 13 |
| advmod:emph | 0 | 0 | 0 | 5 |
| flat:foreign | 0 | 0 | 0 | 4 |
| vocative | 0 | 0 | 0 | 3 |
| compound | 0 | 0 | 0 | 1 |
| **macro avg** | 0.52 | 0.48 | 0.47 | |
| **weighted avg** | 0.68 | 0.71 | 0.68 | |

Table 6: DEPREL evaluation results using the fastText embedding method (averaging embeddings of five words).

| Class | Precision | Recall | F1 score | Support |
|---|---|---|---|---|
| punct | 0.54 | 0.71 | 0.61 | 3037 |
| nmod | 0.42 | 0.35 | 0.39 | 2437 |
| case | 0.28 | 0.56 | 0.37 | 2364 |
| amod | 0.3 | 0.4 | 0.34 | 2355 |
| nsubj | 0.3 | 0.3 | 0.3 | 1725 |
| obl | 0.29 | 0.03 | 0.06 | 1607 |
| root | 0.26 | 0.17 | 0.21 | 1136 |
| conj | 0.27 | 0.38 | 0.32 | 1134 |
| obj | 0.26 | 0.12 | 0.16 | 1072 |
| aux | 0.35 | 0.34 | 0.35 | 1037 |
| cc | 0.22 | 0.25 | 0.24 | 887 |
| advmod | 0.29 | 0.24 | 0.26 | 825 |
| flat | 0.44 | 0.43 | 0.44 | 689 |
| mark | 0.33 | 0.27 | 0.3 | 471 |
| acl | 0.23 | 0.07 | 0.11 | 452 |
| cop | 0.22 | 0.14 | 0.17 | 415 |
| det | 0.25 | 0.13 | 0.17 | 400 |
| xcomp | 0.35 | 0.15 | 0.21 | 350 |
| expl | 0.26 | 0.35 | 0.3 | 302 |
| parataxis | 0.76 | 0.09 | 0.16 | 300 |
| ccomp | 0 | 0 | 0 | 230 |
| discourse | 0.67 | 0.03 | 0.06 | 208 |
| advcl | 0.14 | 0.01 | 0.01 | 198 |
| nummod:gov | 0.27 | 0.6 | 0.37 | 187 |
| appos | 0.2 | 0.02 | 0.04 | 130 |
| nummod | 0.24 | 0.13 | 0.17 | 117 |
| fixed | 0.31 | 0.24 | 0.27 | 100 |
| csubj | 0 | 0 | 0 | 40 |
| det:numgov | 0 | 0 | 0 | 29 |
| orphan | 0 | 0 | 0 | 13 |
| advmod:emph | 0 | 0 | 0 | 5 |
| flat:foreign | 0 | 0 | 0 | 4 |
| vocative | 0 | 0 | 0 | 3 |
| compound | 0 | 0 | 0 | 1 |
| **macro avg** | 0.25 | 0.19 | 0.19 | |
| **weighted avg** | 0.34 | 0.34 | 0.31 | |

Table 7: DEPREL evaluation results using the PyRv+FT embedding method (composing embeddings of five words).

| Class | Precision | Recall | F1 score | Support |
|---|---|---|---|---|
| punct | 1 | 1 | 1 | 3037 |
| nmod | 0.74 | 0.71 | 0.73 | 2437 |
| case | 0.98 | 0.97 | 0.97 | 2364 |
| amod | 0.81 | 0.82 | 0.82 | 2355 |
| nsubj | 0.7 | 0.7 | 0.7 | 1725 |
| obl | 0.59 | 0.63 | 0.61 | 1607 |
| root | 0.61 | 0.69 | 0.65 | 1136 |
| conj | 0.66 | 0.62 | 0.64 | 1134 |
| obj | 0.52 | 0.65 | 0.58 | 1072 |
| aux | 0.78 | 0.94 | 0.85 | 1037 |
| cc | 0.91 | 0.95 | 0.93 | 887 |
| advmod | 0.68 | 0.82 | 0.74 | 825 |
| flat | 0.77 | 0.69 | 0.73 | 689 |
| mark | 0.91 | 0.91 | 0.91 | 471 |
| acl | 0.6 | 0.47 | 0.53 | 452 |
| cop | 0.69 | 0.39 | 0.5 | 415 |
| det | 0.72 | 0.69 | 0.71 | 400 |
| xcomp | 0.67 | 0.59 | 0.63 | 350 |
| expl | 0.86 | 0.99 | 0.92 | 302 |
| parataxis | 0.83 | 0.56 | 0.67 | 300 |
| ccomp | 0.41 | 0.16 | 0.23 | 230 |
| discourse | 0.65 | 0.52 | 0.58 | 208 |
| advcl | 0.38 | 0.22 | 0.28 | 198 |
| nummod:gov | 0.82 | 0.76 | 0.79 | 187 |
| appos | 0.42 | 0.38 | 0.4 | 130 |
| nummod | 0.67 | 0.76 | 0.71 | 117 |
| fixed | 0.67 | 0.61 | 0.64 | 100 |
| csubj | 0 | 0 | 0 | 40 |
| det:numgov | 0.5 | 0.52 | 0.51 | 29 |
| orphan | 0 | 0 | 0 | 13 |
| advmod:emph | 0 | 0 | 0 | 5 |
| flat:foreign | 0 | 0 | 0 | 4 |
| vocative | 0 | 0 | 0 | 3 |
| compound | 0 | 0 | 0 | 1 |
| **macro avg** | 0.58 | 0.55 | 0.56 | |
| **weighted avg** | 0.77 | 0.77 | 0.76 | |

Table 8: Phrases by areas (A, B, and C) in the PyRv+FT representation space (original Croatian phrases).

| Preposition "na" (eng. "on") | | |
|---|---|---|
| Area C | Area B | Area A |
| primjena na | primjena na svijet | na svijet |
| su na | su na lokalnim | na lokalnim |
| asistentent na | asistentent na predmetu | na predmetu |
| medije na | medije na prosvjed | na prosvjed |
| pdv-a na | pdv-a na ulaznice | na ulaznice |
| baziranim na | baziranim na podacima | na podacima |
| financija na | financija na prihodima | na prihodima |
| os na | os na koji | na koji |
| dolara na | dolara na google | na google |
| nalazi na | nalazi na trećoj | na trećoj |
| odnosu na | odnosu na prošlu | na prošlu |
| **Preposition "u" (eng. "in")** | | |
| **Area C** | **Area B** | **Area A** |
| entuzijasta u | entuzijasta u rijeci | u rijeci |
| trenutno u | trenutno u testiranju | u testiranju |
| puštena u | puštena u opticaj | u opticaj |
| opticaj u | opticaj u lipnju | u lipnju |
| aktivnošću u | aktivnošću u nastavi | u nastavi |
| rada u | rada u drvnoj | u drvnoj |
| ulaznice u | ulaznice u europi | u europi |
| glazbenika u | glazbenika u hrvatskoj | u hrvatskoj |
| samo u | samo u protekle | u protekle |
| kap u | kap u moru | u moru |
| se u | se u protekloj | u protekloj |
| godini u | godini u hrvatskoj | u hrvatskoj |
| posto u | posto u odnosu | u odnosu |
| i u | i u prosjeku | u prosjeku |
| . u | . u ovoj | u ovoj |

Table 9: Phrases in area D in the PyRv+FT representation space (original Croatian phrases).

| Area D | |
|---|---|
| **Preposition "na" (eng. "on")** | **Preposition "u" (eng. "in")** |
| na svijet oko | u rijeci koji |
| na lokalnim izborima | u testiranju i |
| na predmetu organizacije | u opticaj u |
| na prosvjed glazbanika | u lipnju 2009. |
| na ulaznice u | u nastavi 1982. |
| na podacima prikupljenim | u drvnoj industriji |
| na prihodima proračuna | u europi . |
| na koji ova | u hrvatskoj samo |
| na google play | u protekle dvije |
| na trećoj poziciji | u moru državnog |
| na prošlu godinu | u protekloj godini |
| | u hrvatskoj nije |
| | u odnosu na |
| | u prosjeku troši |
| | u ovoj hvale |

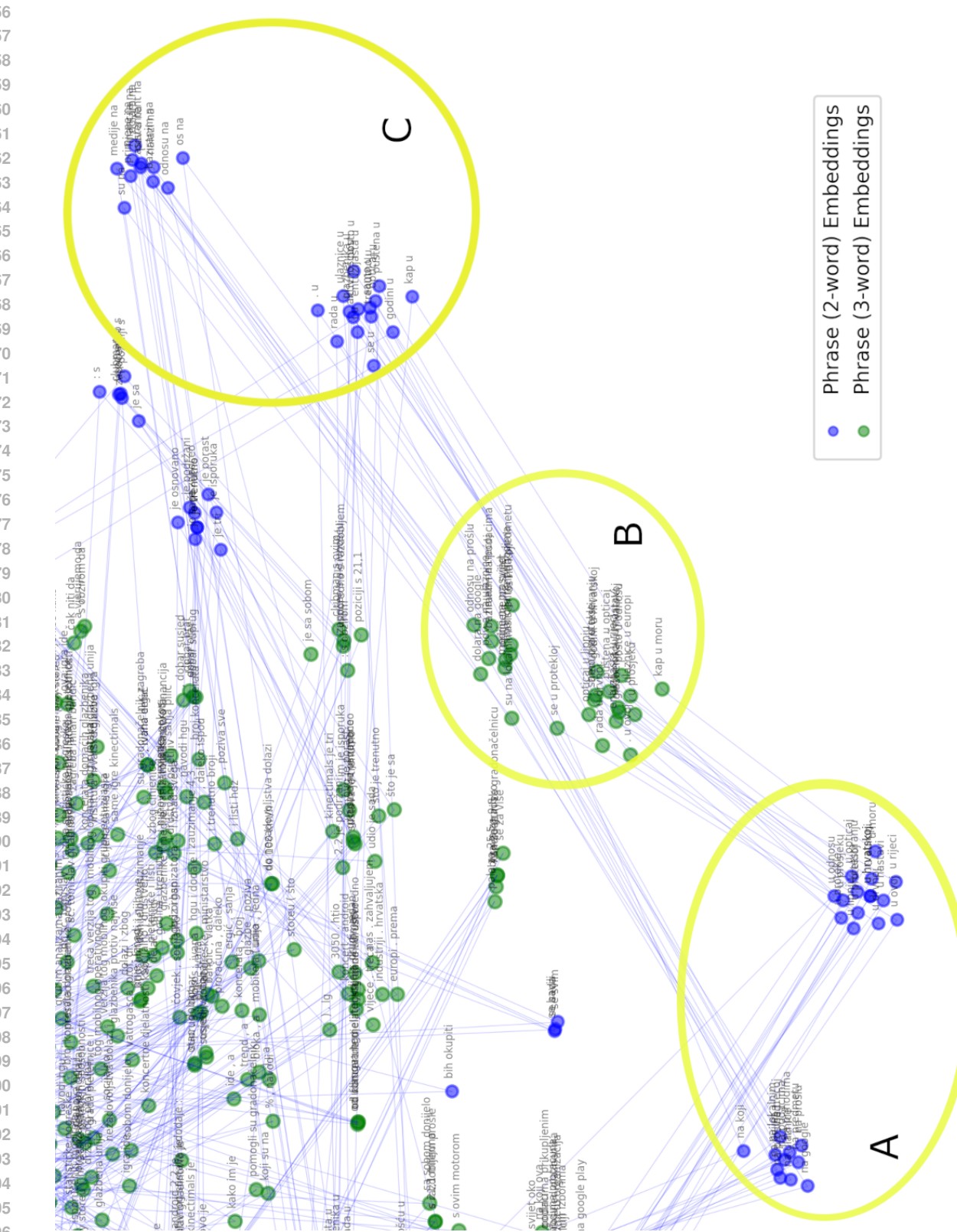

Figure 4: Visualization of the PyRv+FT representation space (reduced from 300 dimensions using t-SNE). Highlighted areas A, B, and C contain phrases with the prepositions 'na' (eng. 'on') and 'u' (eng. 'in'). Tables 8 and 9 (translated: 3 and 4) provide detailed examples of these phrases and their connections within the space.

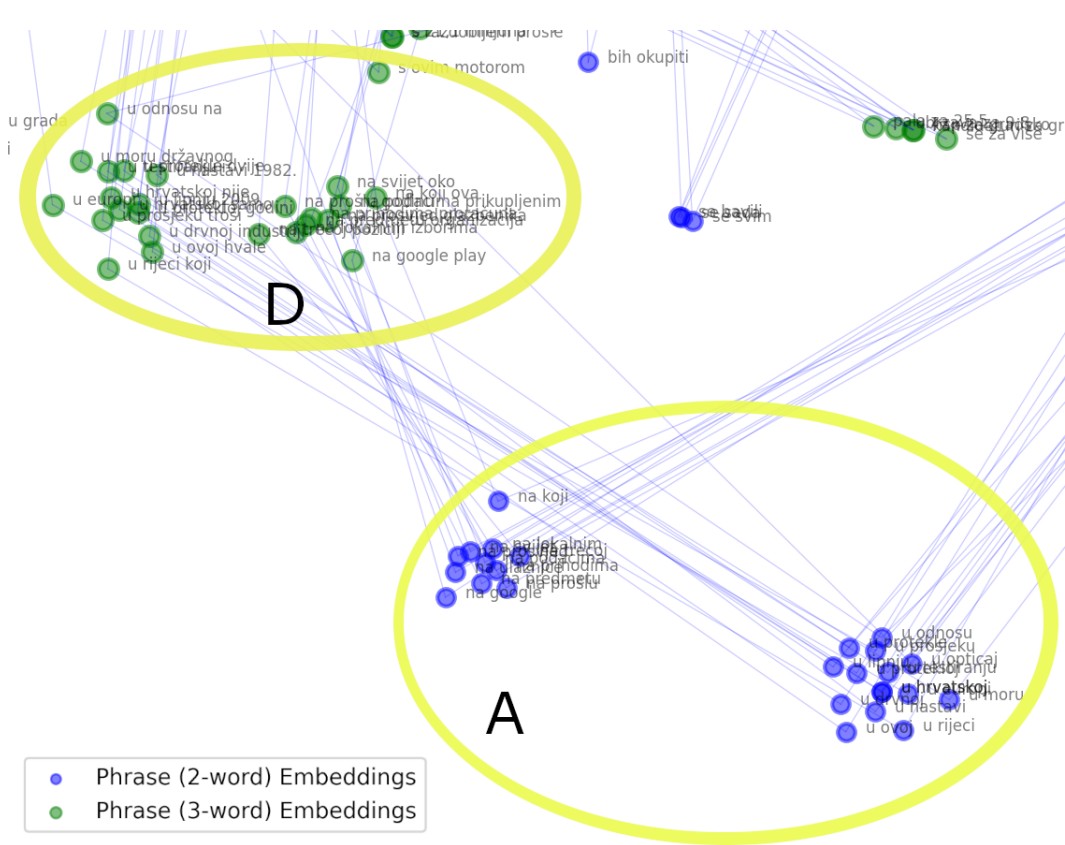

Figure 5: Visualization of the PyRv+FT representation space (reduced from 300 dimensions using t-SNE). Highlighted areas A and D contain phrases structured around the prepositions 'na' (eng. 'on') and 'u' (eng. 'in'). Tables 8 and 9 (translated: 3 and 4) present examples of these phrases and their relationships in the embedding space.

