# OpenReview forum: "Pyramidal Recursive Composition of Multi-Word Units into Unified Representations"
_ICLR.cc/2025/Conference — ICLR 2025 Conference Withdrawn Submission_

### Official Review · Reviewer_WUnK · 2024-10-27

**Soundness:** 3
**Presentation:** 2
**Contribution:** 1
**Rating:** 3
**Confidence:** 4

**Summary:**

The paper explores the performance of PyRv (2024) for word embedding composition, with the difference initial units are word embeddings instead of subword units. The method performs better than naive composition methods on POS tagging and dependency relation.

**Strengths:**

Empirical results on various ways to compose FastText word embeddings for UPOS and DEPREL.

**Weaknesses:**

The paper is not novel in light of the fact that it mostly just uses PyRv. The task itself (how to compose word embeddings) is of limited scope and unlikely to be of broad interest to the ICLR community.

**Questions:**

N/A

---

### Official Review · Reviewer_Wggo · 2024-10-28

**Soundness:** 1
**Presentation:** 3
**Contribution:** 1
**Rating:** 1
**Confidence:** 4

**Summary:**

This paper provides a method for combining word embeddings in a recursive manner, such that embeddings for multi-word units remain meaningful. This is achieved by starting from per-token embeddings, with a pyramid style stacking (akin to a 1d convolution), applied repeatedly. The authors show that pre-training with their architecture results in embeddings that provide an overall better signal for part-of-speech and dependency relation prediction than baseline embedding techniques.

**Strengths:**

The paper is overall clearly written (modulo the training of PyRv, see below), the problem is well motivated, and experimental results presented in an accessible manner.

**Weaknesses:**

The paper has several shortcomings:
1. The approach is not novel. It looks to be a variant of a CNN, with a specific kind of convolution operator - see e.g., Convolutional Neural Networks for Sentence Classification by Yoon Kim (2014), A Convolutional Neural Network for Modelling Sentences by Nal Kalchbrenner, Edward Grefenstette, Phil Blunsom (2014); among many other references. These are just the first papers on CNNs for text, there are dozens more, yet none are cited in the paper. Beyond this, the similarities and difference's to Socher's 2013 work on recursive models needs further elaboration.
1. Word embedding research was very popular around 2010-2015, and this work might have stood to make a contribution back then. However, the field has moved to large language models, and there would need to be a convincing argument for returning to embedding models now. For the Croatian UPOS & DEPREL tasks considered, are these tasks where LLMs do poorly? Alternatively, can any ideas behind your method be used in the context of LLM architectures?
1. The approach is not fully explained. What parameters does the model have? Figures 1 and 2 illustrate the architecture, but purely in a visual way, without reference to composition functions, their parameters etc.  Assuming it has parameters, what loss is used for training (next word prediction?) This is not mentioned anywhere in the paper.
1. Baselines are weak: what about max-pooling instead of averaging, taking the last fast-text token concatenated with the pooled N-1 prior token, and so forth. I expect some variants would at least preserve the 1 word performance of fast-text.
1. Only evaluating on Croatian datasets is odd. It's great to expand the reach of language tools to new corpora, but it's also important to benchmark on existing datasets to demonstrate how general the method is to other languages (the multi-word representation problem is not unique to one language.)
1. Croatian phrases in Tables 3, 4 etc should be presented in the original language as well as translated to English.

**Questions:**

See weaknesses, particular points 1 (novelty) & 2 (understanding the method).

---

### Official Review · Reviewer_WZzx · 2024-11-01

**Soundness:** 1
**Presentation:** 1
**Contribution:** 1
**Rating:** 1
**Confidence:** 5

**Summary:**

The paper proposes a method to compute multi-word embeddings. First, words are mapped to vectors using fastText. Then, their meanings are combined using a "pyramid recursive composition" model, which recursively combines all pair of components in a bottom-up manner until the whole text span is combined. The paper shows that this method substantially outperforms word-embedding-average method (averaging over 5 word embeddings). Compared against the fastText-1-word embedding on UPOS and DEPREL datasets, the results are mixed.

**Strengths:**

None

**Weaknesses:**

The paper hardly meets the ICLR standard. The paper misses related works that employ recursive models without the need for syntactic structures, such as [1,2,3]. The paper also doesn't compare the proposed model against any those baselines. The experimental results are mixed and don't show clear improvement over the very simple baseline fastTest-1-word.

## References


1. Cho, Kyunghyun. "On the Properties of Neural Machine Translation: Encoder-decoder Approaches." arXiv preprint arXiv:1409.1259 (2014).
2. Le, Phong, and Willem Zuidema. "The forest convolutional network: Compositional distributional semantics with a neural chart and without binarization." Proceedings of the 2015 Conference on Empirical Methods in Natural Language Processing. 2015.
3. Drozdov, Andrew, et al. "Unsupervised latent tree induction with deep inside-outside recursive autoencoders." arXiv preprint arXiv:1904.02142 (2019).

**Questions:**

None

---

### Note · Authors · 2024-11-26

I have read and agree with the venue's withdrawal policy on behalf of myself and my co-authors.